# AriEL: volume coding for sentence generation

## Abstract

Saving sequences of data to a point in a continuous space makes it difficult to retrieve them via random sampling. Mapping the input to a volume makes it easier, which is the strategy followed by Variational Autoencoders. However optimizing for prediction and for smoothness, forces them to trade-off between the two. We analyze the ability of standard deep learning techniques to generate sentences through representation sampling. We propose AriEL, an entropic coding method to construct volumes without the need for extra loss terms, and compare those standard learning techniques with its use of the latent space. We benchmark on a toy grammar, to automatically evaluate the language learned and generated, and find where it is stored in the latent space. Then, we benchmark on a dataset of human dialogues and using GPT-2 inside AriEL. Our results indicate that the random access to stored information can be improved, since AriEL is able to generate a wider variety of correct language by randomly sampling the latent space. This supports the hypothesis that encoding information into volumes, improves retrieval of learned information with random sampling.

## 1 Introduction

Representation regularization, through the normalization and bounding of data, representations and gradients, is fundamental to fast deep learning training (Ioffe and Szegedy, 2015; Kingma and Welling, 2014; He et al., 2015; Perez et al., 2018). However, it seldom offers guarantees for boundedness, only encouraging it through initial conditions and loss summands. The final conditions, i.e. the representations learned, can be empirically explored through the sampling of the latent space. This exercise reveals how often the data not seen during training has no bounded representation, which can be regarded as undesirable if we want architectures that can quickly generalize outside the training bias for successful transfer learning. However it is difficult to find the learned patterns through latent sampling, since typically neural networks map an input to a point in $\mathbb{R}^d$ (Hochreiter and Schmidhuber, 1997; Vaswani et al., 2017; LeCun et al., 1989).

Some models do map inputs to volumes, to ease retrieval through random sampling. Variational Autoencoders (Kingma and Welling, 2014; Bowman et al., 2016; Chen et al., 2018) encourage volume representations: by encoding an input into a probability distribution that is sampled before decoding, neighbouring points in $\mathbb{R}^d$ can end up representing one input. However, it requires two loss summands, a log-prior and a log-likelihood, that fight for two different causes. A smooth and volumetric representation, encouraged by the log-prior regularization, can worsen performance, encouraged by the log-likelihood.

By giving partially up on smoothness, we propose AriEL, a method to construct volumes, without a loss to encourage them. It maps sentences to volumes in $\mathbb{R}^d$ for efficient retrieval with random sampling, or a network that operates in its continuous space. It fuses arithmetic coding (AC) (Elias and Abramson, 1963) and k-d trees (KdT) (Bentley, 1975), so we name it *Arithmetic coding and k-d trEes for Language* (AriEL). For simplicity we focus on dialogue language, even if AriEL works with any variable length sequence of symbols. AriEL can be used as a benchmark to understand natural language processing and generation models use of latent space. It fills completely the latent space with the language learned, using

information theory, and bounding its representations within $[0,1]^d$. Its language model splits the latent space in volumes, guided by the probability assigned to the next symbol in a sentence. It can provide an agent with a simpler interface with a pretrained language model, e.g. a GPT-2 (Radford et al., 2019; Wolf et al., 2020), where the agent could choose the optimal $d$. We prove how such a volume representation eases the retrieval of stored learned patterns and how to use it to set references for other models.

Our contributions are therefore:

- AriEL, a novel unsupervised volume representation based on arithmetic coding and k-d trees (Section 3.1), to retrieve learned patterns with random sampling;

- the HouseQ dataset, consisting of a large context-free grammar and a random bias (Section 3.3), to automatically generate and evaluate sentences generated by trained models, and find them in their latent space;

- the notion that explicit volume coding (Section 4) can be a useful technique in tasks that involve the generation of sequences of discrete symbols, such as sentences;

- the observation that conventional learned codes like AE, VAE or Transformer, do not use the latent space effectively (Section 4), in the AriEL entropic coding sense.

## 2 RELATED WORK

**Volume codes:** We define a *volume code* as two functions, an encoder and a decoder functions, where the encoder maps an input $x$ into a set that contains compact and connected sets of $\mathbb{R}^d$ (Munkres, 2018), and the decoder maps every point within that set back to $x$. It is a distributed representation (Hinton et al., 1984) since the input $x$ is represented by at least one $\mathbb{R}^d$ point. We call the volume code *implicit*, when the volumes are encouraged through a loss term (Bengio et al., 2013; Ng and Jordan, 2002; Kingma and Welling, 2014; Jebara, 2012) and *explicit*, when the volumes are constructed through the model's operations, independently from any loss and optimizer choice.

**Sentence generation through random sampling:** Generative Adversarial Networks (GAN) (Goodfellow et al., 2014) map random samples to a learned generation through a 2-players game procedure. Yu et al. (2017); Kusner and Hernández-Lobato (2016); Scialom et al. (2020) significantly improved GAN performance in text generation. Random sampling the latent space is used as well by Variational Autoencoders (VAE) (Kingma and Welling, 2014), to smooth their representations. Bowman et al. (2016); Yang et al. (2017); Li et al. (2021), and others, have refined their performance for text representation. AriEL can be used as a generator or a discriminator in a GAN, or as an encoder or a decoder in an autoencoder. However it differs from them in the explicit procedure to construct volumes. It fills the entire latent space with the learned patterns, to ease retrieval by uniform sampling.

**Arithmetic coding and neural networks:** AC is one of the most efficient lossless data compression techniques (Elias and Abramson, 1963; Witten et al., 1987). AC assigns a sequence to a segment in $[0,1]$ with length proportional to its frequency. When converted into bits, frequent symbols take less bits than unfrequent. AC is used for neural network compression (Wiedemann et al., 2019) and neural networks are used in AC to perform prediction based compression (Jiang et al., 1993; Pasero and Montuori, 2003; Tatwawadi, 2018). We generalize AC to $\mathbb{R}^d$, to combine its properties with the properties of high-dimensional spaces, neural networks domain.

**K-d trees and neural networks:** KdT (Bentley, 1975) is a data structure for storage that can handle different types of queries efficiently. It is typically used as a fast approximation to k-nearest neighbours in low dimensions (Friedman et al., 1977). It gives a binary label to the data with respect to its median. It moves through the k dimensions of the data and repeats the process. Neural networks are typically used in conjuction with KdT to reduce the dimensionality of the search space, for KdT to be able to perform queries efficiently (Woodbridge et al., 2018; Yin et al., 2017; Vasudevan et al., 2009). We use KdT to make sure that the multidimensional AC uses all the space available.

## 3 Methodology

### 3.1 AriEL: volume coding of language in continuous spaces

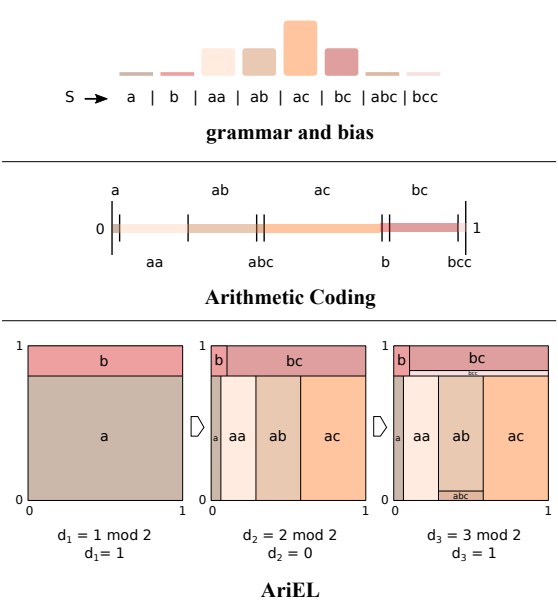

Figure 1: **Arithmetic coding and AriEL.** In this illustrative example, the generating context-free grammar (CFG) is $S \rightarrow a|b|aa|ab|ac|bc|abc|bcc$, and the bar plot on top indicates the frequency of those sentences in the dataset, as an extra bias to the language. Arithmetic Coding (middle) encodes any sequence of this CFG over a single dimension within $[0, 1]$, and the frequency of the sentence determines the length assigned on that segment. AriEL (bottom) is a multidimensional extension of AC (here in 2D), where the frequency information is preserved in the volumes. The Language Model provides the boundaries where the next symbols are to be found. For a 2D latent space, $d = 2$, the axis to split to find symbol $s_t$ is $d_t = t \bmod d$. $d_t = 0, 1$ represent the horizontal and vertical axis.

AriEL maps the sequence/sentence $(s_1, \cdots, s_n) = (s_t)_{t=1}^n$ of length $n$, to a $d$-dimensional volume of size $P((s_1, \cdots, s_n)) = \Pi_{t=1}^n P(s_t|(s_{t'})_{t'<t})$ in the $[0, 1]^d$ hypercube. When the symbol is used as a random variable we refer to it as $s, s'$, while $s_t$ represents the observed sample at time step $t$. The words belong to a finite vocabulary $s \in \{1, \cdots, V_{size}\}, V_{size} \in \mathbb{N}$.

To adapt KdT to more splits than binary, we split axis $d_t = t \bmod d$, into $V_{size}$ segments, one for each possible $s_t$. The segment has length proportional to $s_t$ probability. Then we turn to the following axis $d_{t+1}$, and continue the process of splitting and turning (figure 1 and algorithm 1). In figure 1, $s_t \in \{a, b, c, \_\}$. The initial token $s_1 = a$ is given a portion on $d_1$ of length $P(a)$, larger than the portion given to $s_1 = b$ or $s_1 = c$, since there are less sentences that start with $b$ than with $a$, and there is none that starts with $c$: $P(a) > P(b) > P(c) = 0$. Then, we split $d_2$ according to the probability of the next symbol $s_2$. In this case the second most likely symbol after symbol $s_1 = a$ is $s_2 = c$, so $ac$ ends with a larger volume than $aa$, $ab$ and $a\_$. For sentences longer than $d$, next symbol is assigned an axis $d_3$ previously split, but only the volume selected up to $t - 1$ is further split. So, the sentence $abc$ takes a portion of $ab$ equal to $P(c|(ab))$, while '$ab\_$' takes a portion equal to $P(\_|(ab))$. We estimate language statistics with a Language Model (LM), $P_{LM}(s_t|(s_{t'})_{t'<t})$. This will approximate the frequency information that makes AC entropically efficient.

The sentence is finally encoded as the center of the volume bounded by those segments for simplicity, and any point within it is decoded to the same sentence. The extension to a larger $[a, b]^d$ hypercube is straightforward, and could provide higher precision, but we restrict ourselves to $[0, 1]^d$.

AriEL has a computational complexity of $O(nD^2V_{size})$ for encoding and decoding (algorithm 1), where $n$ is the length of the sequence, $D$ is the dimensionality of the LM latent space, and $V_{size}$ the vocabulary size. AriEL has a minimum number of sequential operations of $O(n)$ for both encoding and decoding, on par with conventional seq2seq recurrent networks.

### 3.2 Neural Networks: models and experimental conditions

We compare AriEL to standard approaches to map variable length discrete spaces to fixed length continuous spaces. These are the sequence to sequence recurrent autoencoders (AE) (Sutskever et al., 2014), their variational version (VAE) (Bowman et al., 2016) and Tran-

**Algorithm 1 AriEL encoder and decoder** $B_{up}$ and $B_{low}$ stand for the upper and lower bounds that define AriEL volumes, main differences between encoder and decoder in blue. The $P_{LM}$ cumulative distributions ($c_{up}$, $c_{low}$) define the volume's limits and size ($a$). $s$ and $s'$ represent the vector of words in the vocabulary, and $s_t$ the observed value at time $t$. (Left) AriEL Encoding: from sentence to continuous space. The volumes are represented by their central point $\mathbf{z}$ for simplicity. (Right) AriEL decoding: from continuous space to sentence. $\mathbf{z}$ is used to identify which volume has to be picked next. Operation $find$ is defined in SM10.

---

| AriEL Encoding | AriEL Decoding |
|---|---|

**AriEL Encoding**

**Input:** sentence: $S = (s_t)_{t=1}^n$
**Output:** $\mathbf{z}$ represents $S$ in $[0, 1]^d$

1: **function** AriEL_ENCODE($S$)
2:     $d$ = latent space dimension
3:     $B_{low} = zeros(d)$
4:     $B_{up} = ones(d)$
5:     $n = length(S)$

6:     **for** $t = 0, \cdots, n-1$ **do**
                  ▷ *choose dimension to split*
7:         $d_t = t \mod d$
8:         $P_{next}(s) = P_{LM}(s|(s_{t'})_{t'<t})$
9:         $c_{low}(s) = \sum_{s>s'} P_{next}(s')$
10:         $c_{up}(s) = \sum_{s>s'-1} P_{next}(s')$
11:         $a = B_{up}(d_t) - B_{low}(d_t)$
                  ▷ *update volume bounds*
12:         $B_{up}(d_t) = B_{low}(d_t) + a \cdot c_{up}(s_t)$
13:         $B_{low}(d_t) = B_{low}(d_t) + a \cdot c_{low}(s_t)$
14:     **end for**
          ▷ *represent the volume by its center*
15:     $\mathbf{z} = (B_{low} + B_{up})/2$
16:     **return z**
17: **end function**

**AriEL Decoding**

**Input:** $\mathbf{z}$ represents $S$ in $[0, 1]^d$
**Output:** sentence: $S = (s_t)_{t=1}^n$

1: **function** AriEL_DECODE($\mathbf{z}$)
2:     $d = dimension(\mathbf{z})$
3:     $B_{low} = zeros(d)$
4:     $B_{up} = ones(d)$

5:     $S = \langle START \rangle$
6:     **for** $t = 0, \cdots, n_{max} - 1$ **do**
                  ▷ *choose dimension to unsplit*
7:         $d_t = t \mod d$
8:         $P_{next}(s) = P_{LM}(s|S)$
9:         $c_{low}(s) = \sum_{s>s'} P_{next}(s')$
10:         $c_{up}(s) = \sum_{s>s'-1} P_{next}(s')$
11:         $a = B_{up}(d_t) - B_{low}(d_t)$
                  ▷ *update volume bounds*
12:         $Bs_{up}(s) = B_{up}(d_i) + a \cdot c_{up}(s)$
13:         $Bs_{low}(s) = B_{low}(d_t) + a \cdot c_{low}(s)$
          ▷ *any point in the volume is assigned the symbol $s_t$*
14:         $s_t = find_s\Big(Bs_{low}(s) < \mathbf{z}(d_t) < Bs_{up}(s)\Big)$
15:         $B_{up}(d_t) = Bs_{up}(s_t)$
16:         $B_{low}(d_t) = Bs_{low}(s_t)$
17:         $S = S.append(s_t)$
18:     **end for**
19:     **return** $S$
20: **end function**

---

former (Vaswani et al., 2017). We trained them for next word prediction, over the biased train set (section 3.3). All can be split into an encoder and a decoder that map the sentences to $\mathbb{R}^d$ and back. More training details in SM7.

In this work, AriEL's language model neural network $P_{LM}$ consists of a word embedding of size 64, a 140-unit LSTM, a feedforward layer and a softmax. At test time the argmax is not applied directly to the softmax, but the latent space is used as the deterministic pointer that chooses the $s_t$, as shown in $find$ definition in SM10. However the LM is trained for next time step prediction through cross-entropy. For AE and VAE, we stack two GRU layers (Cho et al., 2014) with 128 units at both, the encoder and the decoder. The last encoder layer has either $d = 16$ units or $d = 512$ for all methods. The decoder outputs a softmax.

Transformer (Vaswani et al., 2017) is the state-of-the-art in many S2S problems (Dai et al., 2019; Radford et al., 2018). Since at the word level it is a fixed-length representation and it is variable-length at the sentence level, we padded all sentences to the maximum length in the dataset to be able to compare its latent space capacity to the other models. We take as its latent dimension the connection between the encoder and decoder, with $d_{model}$ size, that will take a value of 16 or 512. We choose most parameters as in the original work: the

number of attention heads $n_{head} = 8$, the key and value dimension $d_{key} = d_{value} = 64$, a dropout regularization of 0.1. We change the stack of identical decoders and encoders to $n_{layers} = 2$, and the dimension of the inner feed-forward network to $d_{ff} = 256$ to have a number of parameters similar to the other methods. On the GuessWhat?! dataset (De Vries et al., 2017) we tested $n_{layers} = 20$ to have an amount of parameters comparable for $d = 16$ to the other methods, but performed worse than $n_{layers} = 2$, so we report the smaller one.

All architectures are tested either for next time-step prediction or sampled from the latent space, and all of them are constrained to a small latent space. All of them are 2-layers of encoder and decoder, except for AriEL, since they improved performance compared to when they had only one layer. All of them have about the same number of parameters. All of them could be used as Language Models inside AriEL.

### 3.3 Datasets: toy and human sentences

We perform our analysis on two datasets. A toy dataset of sentences generated from a context-free grammar (CFG) and a realistic dataset of sentences written by humans playing a cooperative game.

**The toy dataset:** we generate questions about objects with a CFG (Supplementary Material 1). To stress the models we choose a CFG with a large vocabulary and numerous grammar rules, rather than classic alternatives (e.g. REBER, Hochreiter and Schmidhuber (1997)). All are questions about house objects, so we call it the HouseQ dataset.

We distinguish between *unbiased* sentences, simply sampled from the CFG, and *biased* sentences, selected according to an additional structural constraint after being sampled from the CFG. To do so we generate an adjacency matrix of words that can occur together in a sentence, and we use that to bias the sentences. Once a sentence is produced from the CFG, if all its words can be together in a sentence judged by the adjacency matrix, the sentence is considered as biased, and unbiased otherwise. For simplicity the adjacency matrix is a random matrix of zeros and ones, generated once for all the experiments, making sure that some symbols like *the*, *it* or *?*, can be found in all sentences. We want to emulate a CFG constrained by realistic scenes, where not all the grammatically correct sentences are semantically correct: e.g. 'Is it the wooden shower in the kitchen ?' could be grammatical, but semantically incorrect since it is unusual. We use it to detect how each learning method extracts the grammar and the roles of each word, despite a bias that makes it harder.

It has a 840 words vocabulary with maximal and mean sentence length of 19 and 9.9 symbols. We split the biased dataset into 1M train, 10k test and 512 validation sentences, with no overlap between them. We created 10k unbiased test sentences with the same CFG, only with sentences that do not follow the adjacency matrix. We train on the biased sentences and we test if they grasped the grammar behind, with the unbiased.

**The real dataset:** we choose the GuessWhat?! dataset (De Vries et al., 2017), a dataset of questions asked by humans to solve a cooperative game. It has a vocabulary of 10,469 words. The maximal and mean length of the sentences are of 57 and 5.9 symbols.

### 3.4 Evaluation Metrics

#### 3.4.1 Quantitative evaluations on the toy grammar, HouseQ

Metrics chosen take advantage of the possibilities opened by the CFG and the random bias, that allow to automatically check the grammatical and bias correctness of the language produced. Standard metrics like perplexity, only check if the word predicted matches in the exact time step the word in a reference sentence. In our view, it makes it an obscure metric for language quality, used in the lack of a better one for (non CFG) human language. We measure generation, prediction and generalization quality. We study networks with a latent dimension of 16 units, to understand their compression limits, and for 512 units, often taken as the default size (Kingma and Welling, 2014; Vaswani et al., 2017).

**Generation/Decoding Quality** is evaluated with sentences produced randomly sampling the latent space of each decoder. The sampling is done uniformly within the maximal hyper-

cube defined by the encoded biased test sentences. We sample 10k sentences and evaluate: *i) vocabulary coverage (VC)* as the ratio of sampled words, over the size of the complete vocabulary; *ii) uniqueness (U)* as a ratio of unique sampled sentences; and *iii) validity (V)* as a ratio of valid sampled sentences, defined as the unique and grammatically correct sentences, the most important of our metrics.

**Prediction Quality** is evaluated by encoding and decoding the 10k *biased* test sentences as follows: *i) prediction accuracy biased (PAB)* as a ratio of correctly reconstructed sentences (i.e. all words must match); *ii) grammar accuracy (GA)* as a ratio of grammatically correct reconstructions (i.e. can be parsed by the CFG, even if the reconstruction is not accurate). and *iii) bias accuracy (BA)* as the ratio of reconstructions that keep the training set bias.

**Generalization Quality** is evaluated using the 10k *unbiased* test sentences while the embeddings were trained on the *biased* training set. The *prediction accuracy unbiased (PAU)*, as *PAB*, is the ratio of correctly reconstructed unbiased sentences. It measures how well the latent space generalizes to grammatically correct sentences outside the training bias.

### 3.4.2   Quantitative evaluations on the real dataset, GuessWhat?!

Humans use spontaneously ungrammatical constructions. We therefore quantify the quality of the language learned with two measures: *memorization* and *subjective interpretability*. *Memorization* is the percentage of sentences that are unique and are found in the training data, indicating how easy it is to retrieve the learned information. To measure *subjective interpretability*, we asked 5 surveyees to evaluate the interpretability of the generated sentences. They were assigned 5 sentences for each method and $d$, with each sentence from a different training seed. Samples shown to the surveyees can be seen in SM12.

### 3.4.3   Interpolations within AriEL

AriEL organization of language in volumes changes with $d$, using the same Language Model. In figure 2 we show how many valid sentences can be found on a straight line between two random points in $[0,1]^d$, AriEL's latent space. Valid refers to unique and grammatically correct, as in the toy dataset case. Since the average is taken over the segment and over the random pairs of points, we call this metric *interpolation diversity*. For each $d$ the average is taken over 15 different random sentence pairs, and over 100 interpolation steps on the line that connects them. The Language Model tested is the one trained on the toy grammar.

### 3.4.4   Qualitative evaluations

We show (1) samples of reconstruction via next word prediction of unbiased sentences, to understand the generalization capabilities of different models ($d = 16$, table 3 and S2), (2) generated sentences by uniformly sampling the latent space, after training on the toy dataset, to understand the generation capabilities ($d = 16$, table 4 and S3) (3) samples generated by AriEL with a pretrained GPT2 as its language model, for small latent spaces ($d = 1, 5$, table 5 and S4) and long sentences (100 symbols), to see if the folding process and the floating point precision represent a limitation for AriEL. To avoid cherry picking, we display the first samples produced.

## 4   Results

### 4.1   Quantitative Evaluations

AriEL improves over the rest for all the 7 measures on the toy data (table 1 and figure S2), outperforming them by a large margin for validity, i.e. unique and grammatical sentences generated, the most important of the metrics. Transformer performs remarkably well at not overfitting and it is able to reconstruct biased and unbiased sentences better than the other non-AriEL methods, even under-parameterized ($d = 16$). However, it performs very poorly at generating diverse valid sentences by random sampling. VAE 16 despite the poor generalization to the biased and the unbiased test set, results in the best non-AriEL generator, measured by validity. The conflict between log-prior and log-likelihood, encouraged VAE to

| | | Generation | | | Prediction | | | Generalization |
|---|---|---|---|---|---|---|---|---|
| | **param** | **vocabulary coverage** | **validity** | **uniqueness** | **bias accuracy** | **grammar accuracy** | **prediction accuracy biased** | **prediction accuracy unbiased** |
| *d = 16* | | | | | | | | |
| **AriEL** | 237K | **70.4 ± 0.2%** | **97.6 ± 0.2%** | **99.7 ± 0.1%** | **100.0 ± 0.0%** | **100.0 ± 0.0%** | **100.0 ± 0.0%** | **53.1 ± 0.4%** |
| **Transformer** | 258K | 70.1 ± 0.8% | 4.7 ± 2.7% | 99.1 ± 0.5% | 99.98 ± 0.01% | 99.95 ± 0.02% | 99.92 ± 0.02% | 49.0 ± 0.1% |
| **AE** | 258K | 6.89 ± 0.7% | 11.5 ± 4.2% | 13.9 ± 5.1% | 89.5 ± 2.3% | 98.0 ± 1.7% | 0.0 ± 0.1% | 0.0 ± 0.1% |
| **VAE** | 258K | 11.5 ± 2.6% | 16.0 ± 9.2% | 24.3 ± 14.8% | 85.4 ± 5.2% | 85.1 ± 8.8% | 0.0 ± 0.1% | 0.0 ± 0.1% |
| *d = 512* | | | | | | | | |
| **AriEL** | 237K | **70.2 ± 0.3%** | **97.9 ± 0.2%** | **99.8 ± 0.1%** | **100.0 ± 0.0%** | **100.0 ± 0.0%** | **100.0 ± 0.0%** | **53.2 ± 0.3%** |
| **Transformer** | 9M | 67.3 ± 0.9% | 17.2 ± 6.3% | 87.2 ± 7.5% | **99.99 ± 0.01%** | 99.91 ± 0.03% | 99.86 ± 0.05% | 49.0 ± 0.1% |
| **AE** | 120M | 39.3 ± 6.0% | 21.0 ± 11.8% | 71.8 ± 5.6% | 82.2 ± 3.5% | 86.8 ± 1.3% | 34.7 ± 11.4% | 24.4 ± 6.0% |
| **VAE** | 120M | 28.9 ± 2.4% | 26.5 ± 2.4% | 95.2 ± 3.8% | 73.8 ± 2.2% | 89.5 ± 2.8% | 4.3 ± 3.7% | 4.9 ± 3.6% |

Table 1: **Evaluation of continuous sentence embeddings on the toy dataset (d = 16, 512).** Each experiment is run 5 times. AriEL, achieves almost perfect performance in most metrics, especially in validity, which quantifies how many random samples were decoded into a unique and grammatical sentence. Transformer performed exceptionally, except for validity. All methods improved their performance increasing $d$, particularly in validity, but still achieved less than one third the performance of AriEL. VAE is the second best in validity, supporting the hypothesis, that volume coding facilitates retrieval of information by random sampling.

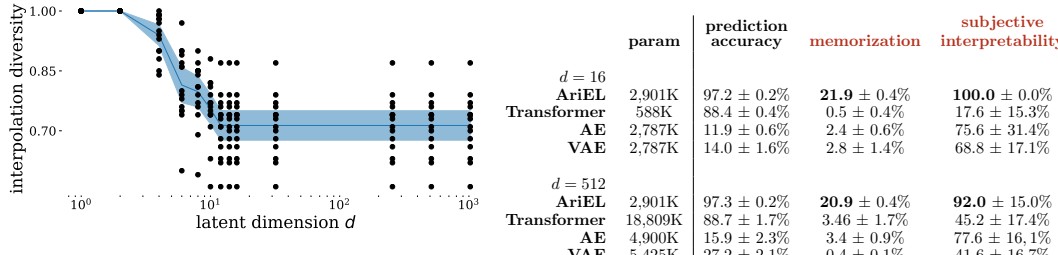

| | **param** | **prediction accuracy** | **memorization** | **subjective interpretability** |
|---|---|---|---|---|
| *d = 16* | | | | |
| **AriEL** | 2,901K | 97.2 ± 0.2% | **21.9 ± 0.4%** | **100.0 ± 0.0%** |
| **Transformer** | 588K | 88.4 ± 0.4% | 0.5 ± 0.4% | 17.6 ± 15.3% |
| **AE** | 2,787K | 11.9 ± 0.6% | 2.4 ± 0.6% | 75.6 ± 31.4% |
| **VAE** | 2,787K | 14.0 ± 1.6% | 2.8 ± 1.4% | 68.8 ± 17.1% |
| *d = 512* | | | | |
| **AriEL** | 2,901K | 97.3 ± 0.2% | **20.9 ± 0.4%** | **92.0 ± 15.0%** |
| **Transformer** | 18,809K | 88.7 ± 1.7% | 3.46 ± 1.7% | 45.2 ± 17.4% |
| **AE** | 4,900K | 15.9 ± 2.3% | 3.4 ± 0.9% | 77.6 ± 16,1% |
| **VAE** | 5,425K | 27.2 ± 2.1% | 0.4 ± 0.1% | 41.6 ± 16.7% |

Figure 2: **Interpolations between random points in AriEL's latent space, and amount of valid sentences generated in between.** For a low latent dimension all sentences are very densely packed, and in the extreme of $d = 1$, all sentences are found on the same axis. As $d$ increases, the sentences are redistributed in $[0, 1]^d$ and less sentences are found in a given direction. The lower bound at 0.746 can change depending on the training dataset.

Table 2: **Performance on the Guess-What?! Questioner data.** For the real dataset the pattern is repeated: AriEL shows that a larger value of valid sentences is possible. Transformer 16 gave better results when $n_{layers} = 2$ than $n_{layers} = 20$, that was tested to increase its learnable parameters from 588K to 2,666K.

look for sentences outside the bias, since it was able to produce more grammatically correct sentences, albeit unbiased, than AE. Increasing the learned parameters ($d = 512$), had no effect on Transformer, that was already excellent in several of the metrics, apart from a significant improvement in validity. However, a larger latent space and the increase in number of parameters that followed, prevented AE and VAE from overfitting (better PAU and PAB). When trained on human sentences, on the GuessWhat?! dataset, AriEL sets again a large memorization and interpretability margin compared to the other approaches. Generated samples can be found in table S5.

In the interpolation diversity study (figure 2) we see that for low $d$, we have to pass through many sentences in between two random points in the latent space, while as we augment the dimensionality, we distribute the sentences in different directions. Therefore we find less sentences when we move on a straight line between two random points.

**Input Sentences**
is it huge and teal ?
is the thing transparent , huge and slightly heavy ?
**AriEL**
is it huge and teachable ?
is the thing transparent , huge and slightly heavy ?
**Transformer**
is it huge and magenta ?
is the thing transparent , huge and slightly heavy ?
**AE**
is it this average-sized and average-sized laminate ?
is the thing very heavy , heavy and very heavy ?
**VAE**
is it the light deep bedroom ?
is the thing textured , textured and moderately heavy ?

Table 3: **Generalization: next word prediction of unbiased sentences at test time.** Blue means that the incorrect reconstruction complies with the bias and purple means that it is still unbiased. Most reconstructions seem grammatically correct. In practice AriEL also made errors. Some failed reconstructions comply with the training bias, some do not. Interestingly the errors made by Transformer tend to turn the unbiased input sentence into a biased version. AE produced only biased sentences whose structure resembled the unbiased ones. VAE behaved similarly, producing more unbiased sentences.

**AriEL**
is the object that tiny very light set ?
is the thing a tiny destroyable abstraction ?
**Transformer**
is it an tomato slot box made of decoration facing stone ?
is the thing short and spring heavy slightly heavy potang ?
**AE**
is the object that light light laminate ?
is the thing a light , small and small laminate ?
**VAE**
is the thing a light and deep office ?
is it light , light and light and pink ?

Table 4: **Generation: output of the decoder when sampled uniformly in the latent space.** Red defines grammatically incorrect generations according to the CFG the models are trained on. AriEL produces an extremely varied set of grammatically correct sentences, most of which keep the bias of the training set. Transformer reveals itself to be hard to control via random sampling of the latent space, since it almost never produces correct sentences with this method. AE and VAE manage to produce several different sentences, the latter producing more non grammatical, but as well more varied grammatical ones.

## 4.2 Qualitative Evaluations

Table 3 and S2 show the generalization study. AE and VAE fail to generalize to the unbiased language, but both keep the structure of the input sentence. Their behavior improved with $d = 512$, and its increase of parameters. In theory, AriEL can reconstruct any sequence by design. In practice, it failed slightly less often than Transformer. Both produce reconstructions of the unbiased input at a similar rate (table 3, 1 and figure S2). This means that codes for data not seen during training are available for AriEL and Transformer. Instead, the latent space seems to be taken exclusively by the training set for AE and VAE, since sentences that are not seen during training (e.g. unbiased) cannot be reconstructed.

A very high number ($97.6 \pm 0.2\%$) of AriEL's generations are *valid* (table 4 and 1). AE and VAE perform well despite the small latent space. VAE triples AE in *validity* when $d = 16$ (table 1). Transformer struggles to generate grammatical sentences sampling the latent space. However, increasing $d$, Transformer, AE and VAE improve in validity, remaining at one third AriEL's reference.

Samples generated with GPT-2 as AriEL's LM can be seen in table 5 and S4, for $d = 1, 5, 10, 50, 100$. Despite the large vocabulary of GPT-2, 50K subwords, and the long sentences, 100 symbols, the floating point precision and the unit hypercube latent space of AriEL, seem to still be able to generate high quality language.

## 5 Discussion

**AriEL latent space $d$ is a free parameter.** The size $d$ of the latent space of AriEL can be defined at any time, for a fixed Language Model. It could therefore be controlled with a learnable parameter or with the activity of another neuron. As we increase $d$, the volumes will have more neighbouring volumes that represent different sentences (figure 2).

---

**$d = 1$**

---

Where to go from here?$\backslash n \backslash n$We're looking for special interests and fans to pay much more attention to when they come in to another site. If you're looking for good stuff to read on other sites, then get on your site and read their stories. If you're looking for things that work for your community, then look up what their leaderboard is and where they're based. If you're looking for things that make me love them, then say hey. You're really doing your

Saul Niguez is not going to let go of his dream of playing for Hillary Clinton.$\backslash n \backslash n$The Florida native will plan to accept the Democratic presidential nomination for president on November 8.$\backslash n \backslash n$The 38-year-old machinist has called Trump$\backslash$'s presidential nominee a "post-modern bull - " on social media.$\backslash n \backslash n$In an interview with DailyMail.com, Niguez told her you could still name her after Trump.$\backslash n \backslash n$"This is my father$\backslash$'s

---

**$d = 5$**

---

Utility Shells are just as dependent on human labor as steel, yet they had been enormously more efficient than nuclear power in incongruity's late 70s. US clients – including the nation's largest retiree retirement fund – already manufactured many of their own carbon-free cars. But what about non-node Americans? How could they simply cool down at home for work for years at a time? Inventing these subs would seem like a smart thing to do. But there's one

Out featured: on Alcoholism, Marriage and Dating: I'am No Timber [Decision note: 19th August 2013] with Runnymede Colman and producer Dan Patrick$\backslash n \backslash n$Partial transcript as follows:$\backslash n \backslash n$DAVID WEISES: After 20 years of marriage to my Malcastian partner, we've lost our lives to drugs. And so it's our hope that by allowing my Malcastian partner to become an alcoholic and grow up to be a successful husband and

Table 5: **AriEL samples using GPT-2 as Language Model.** Even for a large language model such as the small GPT-2, 117M parameters, over 50K subwords, and 100 symbols sentences, AriEL results in high quality samples for a wide range of latent dimensions (SM 11). Floating point precision does not seem to be of concern up to this limit.

**Evidence for volume codes.** The results suggest that AriEL volumes are responsible of its success. We provided evidence on how volume codes improve the retrieval of information composed of discrete, variable length sequences, by random sampling, compared to other codes. AriEL generates more valid sentences, an explicit volume code. VAE is the second on the toy dataset, an implicit volume code, but it is worse than AE on the real dataset.

**Transformers are hard to sample from the latent space.** The Transformer has been used in this work in an uncommon way: by sampling its latent space instead of its output space. Its low validity score in this work reflects it. Our aim was to understand the latent organization of language, so, we do not suggest this is the most effective way to use Transformer. Transformer is excellent when sampled in the output space, but it's difficult to sample from the latent space. This is so because Transformer represents each word in $d$ dimensions while the other approaches represent a sentence in $d$ dimensions. Transformer needs a very high dimensional vector to represent a sentence, $n \cdot d$, where $n$ is the number of words in it. This makes it hard to find sentences by uniform sampling the latent space.

## 6 CONCLUSIONS

We proposed AriEL, a mapping of language into volumes, that we used as a reference system for language generation. AriEL fuses arithmetic coding and k-d trees to construct volumes that preserve the statistics of a dataset. On the one hand, this study helps to realize how much of the latent space lies unused by standard architectures. On the other hand, when compared to standard techniques it highlights room for improvement in their capacity for generation, prediction and generalization. AriEL allows us to sample/generate in theory the training set probability distribution and in practice a diverse set of sentences, as demonstrated on the toy, on the human dataset and using GPT-2 as its language model.

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

SUPPLEMENTARY MATERIAL

# 1 CONTEXT-FREE GRAMMAR (CFG) USED IN THE EXPERIMENTS

The context free grammar used to generate the biased and unbiased sentences is composed by the following rules:

```
s  →  q

q  →  qword adjective ',' adjective 'and' adjective '?'
q  →  qword adjective 'and' adjective '?'
q  →  qword adjective '?'
q  →  qword 'made' 'of' noun_material '?'
q  →  qword preposition np '?'
q  →  qword np '?'

np  →  determiner adjective adjective adjective noun
np  →  determiner adjective ',' adjective 'and' adjective noun
np  →  determiner adjective 'and' adjective noun 'made' 'of' noun_material
np  →  determiner adjective adjective noun
np  →  determiner adjective 'and' adjective noun
np  →  determiner adjective noun 'made' 'of' noun_material
np  →  determiner noun 'made' 'of' noun_material
np  →  determiner adjective noun
np  →  determiner noun

qword  →  'is' 'it'  |  'is' 'the' 'object'  |  'is' 'the' 'thing'
noun  →  noun_object  |  noun_material  |  noun_roomtype
preposition  →  preposition_material

adjective  →  adjective_color  |  adjective_affordance  |  adjective_overall_size  |
              adjective_relative_size  |  adjective_relative_per_dimension_size  |
              adjective_mass  |  adjective_state  |  adjective_other

noun_object  →  'accordion'  |  'acoustic' 'gramophone'  |  'bar'  |  'barrier'  |
                'basket'  |  'outdoor' 'lamp'  |  'outdoor' 'seating'  |  ...

noun_material  →  'bricks'  |  'carpet'  |  'decoration' 'stone'  |  'facing' 'stone'  |
                  'grass'  |  'ground'  |  'laminate'  |  'leather'  |  'wood'  |  ...

noun_roomtype  →  'aeration'  |  'balcony'  |  'bathroom'  |  'bedroom'  |  'boiler' 'room'  |
                  'garage'  |  'guest' 'room'  |  'hall'  |  'hallway'  |  'kitchen'  |  ...

determiner  →  'a'  |  'an'  |  'that'  |  'the'  |  'this'

preposition_material  →  'made' 'of'

adjective_color  →  'antique' 'white'  |  'magenta'  |  'maroon'  |
                    'slate' 'gray'  |  'white'  |  'yellow'  |  ...

adjective_affordance  →  'actable'  |  'addable'  |  'addressable'  |  'deliverable'  |
                         'destroyable'  |  'dividable'  |  'movable'  |  ...

adjective_size  →  adjective_overall_size  |  adjective_relative_per_dimension_size

adjective_overall_size  →  'average-sized'  |  'huge'  |  'large'  |  'small'  |  'tiny'
adjective_relative_per_dimension_size  →  'deep'  |  'narrow'  |  'shallow'  |
                                          'short'  |  'tall'  |  'wide'

adjective_mass  →  'heavy'  |  'light'  |  'moderately' 'heavy'  |  'moderately' 'light'  |
                   'slightly' 'heavy'  |  'very' 'heavy'  |  'very' 'light'
adjective_state  →  'closed'  |  'opened'
adjective_other  →  'textured'  |  'transparent'
```

## 2   SIZE OF THE CFG LANGUAGE SPACE

From the CFG used in the experiment, it is possible to extract a total of 15,396 distinct grammar rules, some are shown below. However, for simplicity, we defined only 4, related to the number of adjectives in it. In the case of the unbiased dataset, those rules can produce a total of 9.81e+18 unique sentences. The total number of unique sentences for the biased dataset is expected to be an order of magnitude smaller.

```
[qword,  prep_material,  determiner,  adj_state,  'and',  adj_other,  noun_roomtype,  '?']
[qword,  prep_spatial,  determiner,  adj_other,  adj_state,  adj_state,  noun_object,  '?']
[qword,  determiner,  adj_other,  ',',  adj_mass,  'and',  adj_affordance,  noun_roomtype,  '?']
[qword,  determiner,  adj_relative_per_dimension_size,  adj_overall_size,  noun_object,  '?']
[qword,  determiner,  adj_overall_size,  ',',  adj_state,  'and',  adj_state,  noun_material,  '?']
[qword,  prep_spatial,  determiner,  adj_other,  adj_mass,  adj_affordance,  noun_material,  '?']
[qword,  adj_state,  'and',  adj_relative_size,  '?']
[qword,  prep_material,  determiner,  adj_mass,  adj_other,  adj_other,  noun_material,  '?']
[qword,  prep_spatial,  determiner,  adj_state,  adj_other,  adj_color,  noun_object,  '?']
[qword,  determiner,  adj_relative_size,  'and',  adj_overall_size,  noun_material,  '?']
[qword,  determiner,  adj_state,  adj_overall_size,  adj_other,  noun_roomtype,  '?']
[qword,  determiner,  adj_other,  adj_state,  adj_mass,  noun_material,  '?']
[qword,  determiner,  adj_overall_size,  'and',  adj_other,  noun_material,  '?']
[qword,  determiner,  adj_color,  adj_other,  noun_object,  '?']
[qword,  prep_spatial_rel,  determiner,  adj_mass,  adj_color,  noun_roomtype,  '?']
[qword,  determiner,  adj_state,  'and',  adj_relative_size,  noun_object,  '?']
[qword,  determiner,  adj_color,  adj_color,  adj_relative_size,  noun_material,  '?']
[qword,  determiner,  adj_affordance,  noun_object,  '?']
[qword,  determiner,  adj_other,  adj_other,  adj_state,  noun_roomtype,  '?']
```

## 3   EXAMPLE OF SENTENCES GENERATED FROM THE CFG

### 3.1   BIASED SAMPLE SENTENCES

- is it large, light yellow and light ?
- is it white, deep pink and average-sized ?
- is it a light, huge and shallow laminate ?
- is the object average-sized and light ?
- is the object fashionable, ghost white and pale turquoise ?
- is the thing huge, huge and khaki ?
- is the thing small, ignitable and very light ?
- is the object a notable very light orange carpet ?
- is the object this small wood made of facing stone ?
- is the object a textured and combinable floor cover made of laminate ?

### 3.2   UNBIASED SAMPLE SENTENCES

- is the object the huge tiny lovable guest room ?
- is the object the closed closed transparent textile ?
- is the thing a transparent, narrow and slightly heavy textile ?
- is it steerable, dark orange and light ?
- is it gray, very heavy and textured ?
- is it closed, heavy and moderately light ?
- is it transparent, transformable and moderately light ?
- is the thing average-sized and dark red ?
- is the thing large and deep garage ?
- is it that slightly heavy stucco made of grass ?

## 4  CFG Vocabulary

| Annotation | Nb. of classes | Example of classes |
|---|---|---|
| Noun | 86 | air conditioner, mirror, window, door, piano |
| WordNet category (Miller, 1995) | 580 | instrument, living thing, furniture, decoration |
| Location | 24 | kitchen, bedroom, bathroom, office, hallway, garage |
| Color | 139 | red, royal blue, dark gray, sea shell |
| Color property | 2 | transparent, textured |
| Material | 15 | wood, textile, leather, carpet, decoration stone |
| Overall mass | 7 | light, moderately light, heavy, very heavy |
| Overall size | 4 | tiny, small, large, huge |
| Category-relative size | 10 | tiny, small, large, huge, short, shallow, narrow, wide |
| State | 2 | opened, closed |
| Acoustical capability | 3 | sound, speech, music |
| Affordance | 100 | attach, bend, divide, play, shake, stretch, wear |

Table S1: Description of vocabulary used.

## 5  Use of latent space

In figure S1, each dot represents a sentence in the latent space. In the first row the dot in the latent space is passed as input to the decoder, while in the second and third row the dot is the output of the encoder when the biased test sentence is fed at its input. Two random axis in $\mathbb{R}^d$ are chosen for the generator, first row, while two axis were chosen subjectively among the first components of a PCA for the encoder, second and third row. In every case, the values in the latent space where normalized between zero and one to ease the visualization. Lines are used to ease the visualization of the clusters of data with their label, since the point clouds overlap and are hard to see. The curves are constructed as concave hulls of the dots based on their Delaunay triangulation, a method called alpha shapes Edelsbrunner et al. (1983).

We can see in figure S1 (first row) how easy it is to find grammatical sentences when randomly sampling the latent space for each model. AriEL practically only generates grammatical sentences and AE and VAE perform reasonably well too, while Transformer fails. AriEL failures are plot on top, to remark how few they are, while AE, VAE and Transformer failures are plot at the bottom, otherwise they would hide the rest given how numerous they are. In the same figure (rows two and three) we can observe how different methods structure the input in the latent space, each with prototypical clusters. The Transformer presents an interesting structure of clusters whose purpose remains unclear. Interestingly, the encoding maps seem to be more organized than the decoding ones. All the models seem to cluster data belonging to different classes at the encoding, that could be taken advantage of by a learning agent placed in the latent space. However it seems hard to use the Transformer as a generator module for an agent. The good performance of AriEL is a consequence of the fact that all the latent space is utilized, and in no directions large gaps can be observed. This can be seen in the two encoding rows, where the white spaces around the cloud of dots are consequence of the rotation performed by the PCA, otherwise all the space between 0 and 1 would be utilized by AriEL.

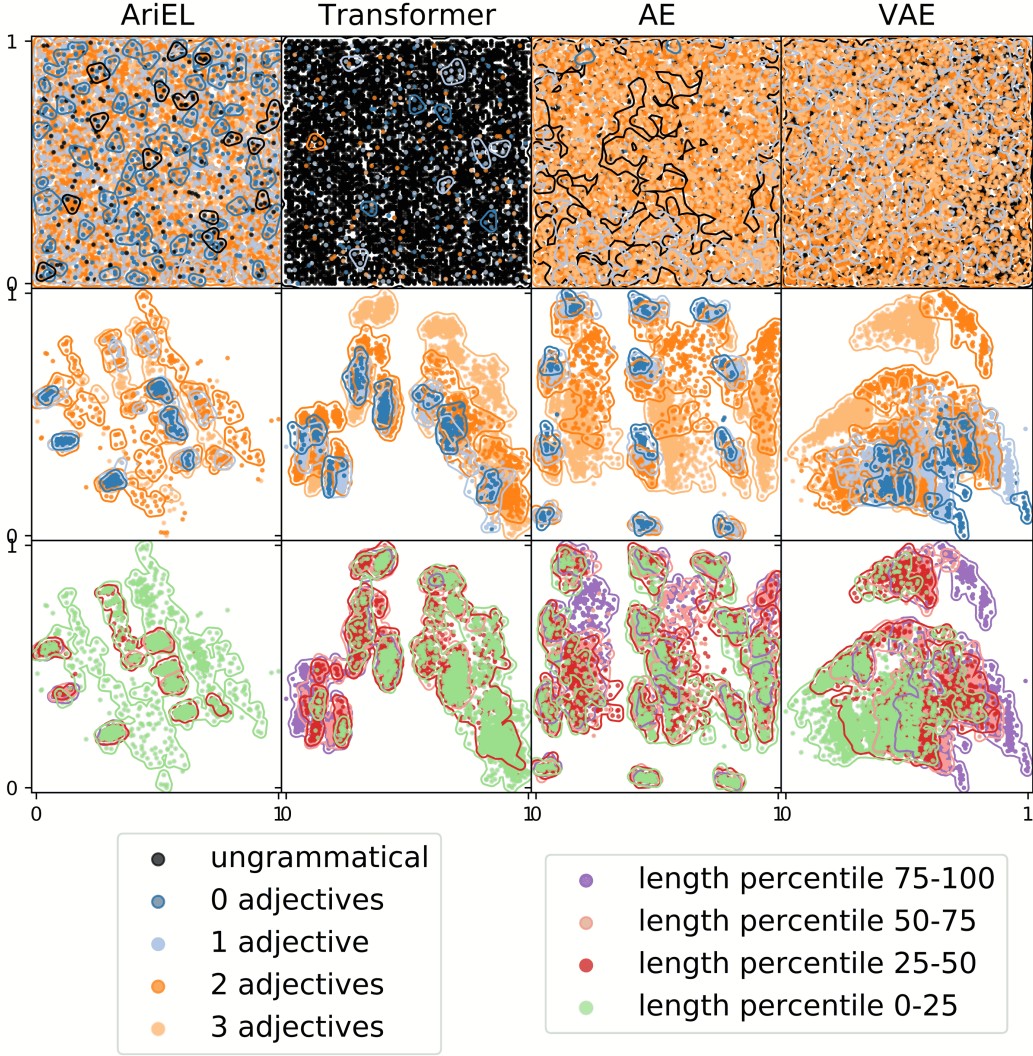

Figure S1: **Random-sampling-based generation in the first row, and encoding of input sentences in the remaining rows.** A sentence is represented by a point in the latent space. First row shows the proportion of grammatically correct sentences that can be decoded by random uniform sampling the latent space. AriEL sampled almost only grammatical sentences (ungrammatical are so few that are placed on top in the plot). Transformer mainly yielded ungrammatical sentences, while AE and VAE were able to produce many grammatical sentences (ungrammatical are below, otherwise they would cover up the grammatical). Each dot is labeled according to how many adjectives the sentence generated has. Second and third rows show the clusters of points in the latent space for the test sentences as they are mapped by the encoders. All models seem to shift the clusters to some degree according to the number of adjectives in the sentence, in the second row. A similar conclusion applies to the third row, that shows where sentences of different length are encoded. For all panels, we searched subjectively for the dimensions that would better reveal some clustering, with the help of PCA. We scaled all latent representations between [0,1] for visualization.

## 6 Visualization of performance on toy data

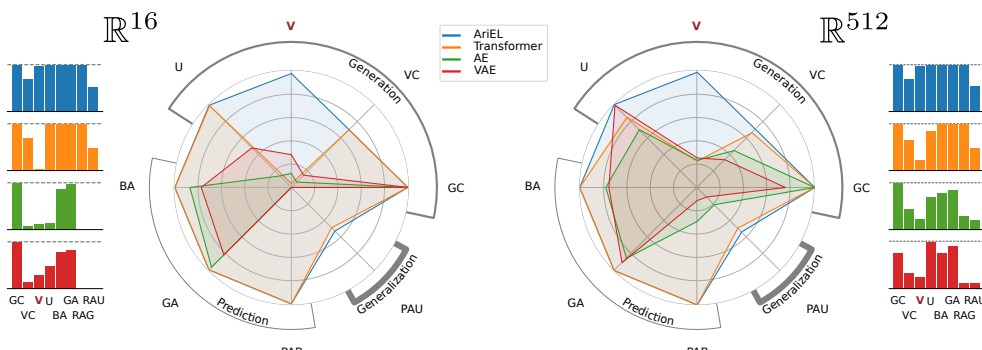

Figure S2: **Radar Chart of the Quantitative Assessment. Latent space of $\mathbb{R}^{16}$ on the left and $\mathbb{R}^{512}$ on the right.** Training was performed on biased sentences. The metrics are defined in Methodology: Generalization is measured by *prediction accuracy of unbiased* sentences (PAU), Prediction by *prediction accuracy of biased* sentences (PAB), *grammar accuracy* (GA) and *bias accuracy* (BA) and Generation by *uniqueness* (U), *validity* (V), *vocabulary coverage* (VC) and *grammar coverage* (GC). AriEL excels in all the 8 metrics. Most importantly AriEL outperforms every other method in Generation Validity (V) and it doesn't require a large latent space to do so ($\mathbb{R}^{16}$ similar to $\mathbb{R}^{512}$). VAE performs remarkably well at generating unique and grammatical sentences (validity, V) when the latent space is small ($\mathbb{R}^{16}$), probably given the volume-code nature of the method. Transformer performs exceptionally at not overfitting in the reconstruction tasks and generalizing, it manages to cover all grammar rules, even with a very small number of parameters ($\mathbb{R}^{16}$). Transformer proved to be an inefficient generator using random sampling as input (validity) but improved with a larger latent space. For a larger latent space of $\mathbb{R}^{512}$, AE and VAE overfit less (PAU and PAB) and improve their Generation (V).

## 7 Training details

We go through the training data 10 times, in mini-batches of 256 sentences. We applied teacher forcing (Williams and Zipser, 1989) during training. We use the Adam (Kingma and Ba, 2015) optimizer with a learning rate of 1e-3 and gradient clipping at 0.5 magnitude. Learning rate was reduced by a factor of 0.2 if the loss function didn't decrease within 5 epochs, with a minimum learning rate of 1e-5. For all RNN-based embeddings, kernel weights used the Xavier uniform initialization (Glorot and Bengio, 2010), while recurrent weights used random orthogonal matrix initialization (Saxe et al., 2014). Biases are initialized to zero. Embeddings layers are initialized with a uniform distribution between [-1, 1]. For Transformer the multihead attention matrices and the feedforward module matrices, used the Xavier uniform initialization (Glorot and Bengio, 2010), the beta of the layer normalization uses zeros, and its gamma uses ones for initialization. AE and VAE are trained with a word dropout of 0.25 at the input, and VAE is trained with KL loss annealing that moves the weight of the KL loss from zero to one during the 7th epoch, similarly to the original work (Bowman et al., 2016). Our code is in TensorFlow (Abadi et al., 2015). We run our experiments on an NVIDIA TITAN Xp and an NVIDIA Tesla K40c. Each experiment took less than one day to converge.

## 8 AriEL is an explicit volume code proof

AriEL uses a bounded region of $\mathbb{R}^d$, the interval $[0, 1]^d$, so encoder and decoder map each input to and from a compact set. Moreover, any sequence $x$ is assigned to a hyper-rectangle (Johnson, 2018) and back. Since hyper-rectangles cannot be divided into two disjoint non-empty closed sets, they are connected (Munkres, 2018). Therefore AriEL is a *volume code*.

AriEL is an *explicit volume code* since its LM is trained only on a next word prediction log-likelihood loss, without a regularization term that encourages smoothness, and the volumes are constructed by arranging the softmax outputs into a $d$ dimensional grid, operation performed with any choice of loss or optimizer.

## 9  MORE SENTENCES FROM THE QUALITATIVE STUDY

**Input Sentences**
is the thing this linen carpet made of tile ?
is it huge and teal ?
is the thing transparent , huge and slightly heavy ?
is the object antique white , tiny and closed ?
**AriEL**
is the thing this lime carpet made of tile ?
is it huge and teachable ?
is the thing transparent , huge and slightly heavy ?
is the object antique white , tiny and closed ?
**Transformer**
is the thing this stretchable carpet made of tile ?
is it huge and magenta ?
is the thing transparent , huge and slightly heavy ?
is the object antique white , tiny and closed ?
**AE**
is the thing this small toilet made of laminate ?
is it this average-sized and average-sized laminate ?
is the thing very heavy , heavy and very heavy ?
is the object light pink , small and textured ?
**VAE**
is the thing a small and textured deep stone ?
is it the light deep bedroom ?
is the thing textured , textured and moderately heavy ?
is the thing light , moderately heavy and light green ?

**AriEL**
is the object that tiny very light set ?
is the thing a tiny destroyable abstraction ?
is the thing this mint cream textured organic structure ?
is the object this small large wearable textile ?
**Transformer**
is the thing slightly heavy heavy stone squeezable
closed sea heavy ?
is it pale lime executable executable shallow decoration
drab turquoise , heavy and potang ?
is it an tomato slot box made of decoration facing stone ?
is the thing short and spring heavy slightly heavy potang ?
**AE**
is the object that light light laminate ?
is the thing a light , small and small laminate ?
is the thing that tiny small decoration stone ?
is the object the average-sized , textured and
average-sized laminate ?
**VAE**
is the thing a light and deep office ?
is it light , light and light and pink ?
is the object dark , light and pink ?
is the object a light deep living room ?

Table S2: **Generalization: next word prediction of unbiased sentences at test time.** An unbiased sentence is encoded and decoded by each model. Color means that the word was incorrectly reconstructed. Blue means that the sentence complies with the bias and purple means that the incorrect reconstruction is still unbiased. Most reconstructions seem grammatically correct. In practice AriEL also made errors. Some of its failed reconstructions comply with the training bias, some do not. Transformer performs remarkably well, and interestingly the errors made tend to turn the unbiased input sentence into a biased version at the output. AE produced only biased sentences whose structure resembled the unbiased ones. VAE behaved similarly, producing more unbiased sentences.

Table S3: **Generation: output of the decoder when sampled uniformly in the latent space.** Red defines grammatically incorrect generations according to the CFG the models are trained on. AriEL produces an extremely varied set of grammatically correct sentences, most of which keep the bias of the training set. Transformer reveals itself to be hard to control via random sampling of the latent space, since it almost never produces correct sentences with this method. AE and VAE manage to produce several different sentences, the latter producing more non grammatical, but as well more varied grammatical ones.

## 10    *find* OPERATION

The find operation defines within which bounds the latent vector is pointing at, and therefore what is the next word selected by AriEL. This operation is performed every time an axis $d_i$ is selected. We define it with an argmax operation at the end, but other solutions can be defined. To give it a pseudonim we call it *DetRoulette*, from *Deterministic Roulette*.

---

**Algorithm 2** find operation

---

**Input:** bounds on axis $d_i$: $Bs_{low}(s)$, $Bs_{up}(s)$ ; value of latent space on $d_i$: $\mathbf{z}(d_i)$
**Output:** $s_i$ is the word $\mathbf{z}$ was pointing at,

1: **function** FIND$_s$($\text{Bs}_{low}(s) < \mathbf{z}(d_i) < Bs_{up}(s)$)
2:     $a(s) = Bs_{up}(s) - \mathbf{z}(d_i)$
3:     $b(s) = Bs_{low}(s) - \mathbf{z}(d_i)$
4:     $logits(s) = -a(s) \cdot b(s)$
5:     $s_i = \text{argmax}_s(logits(s))$
6:     **return** $s_i$
7: **end function**

---

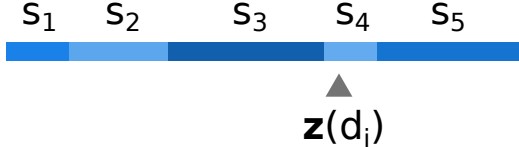

Figure S3: The latent space $\mathbf{z}$ does not necessarily point at the most likely word. Here the $find$ function would return word $s_4$, the least likely. The bounds define the limits of the different blue areas.

## 11   SAMPLES FROM ARIEL WITH GPT-2 AS LANGUAGE MODEL

**$d = 10$**

He must\'ve been very clear that the missive was a sign of trouble since the MTA was lukewarm on keeping maintenance intact. "It only takes a few seconds for you to walk through the lines and see the notice," Kris Johnson, who worked on the 14th Street Bus line in January, said of the notice. "But we don\'t hire people that don\'t want to work as long as they work."\n\nHow many times have you seen "Why $15 is $15

Dear Reader, As you can imagine, more people are reading The Jerusalem Post than ever before. Nevertheless, traditional business models are no longer sustainable and high-quality publications, like ours, are being forced to look for new ways to keep going. Unlike many other news organizations, we have not put up a paywall. We want to keep our journalism open and accessible and be able to keep providing you with news and analyses from the frontlines of Israel, the Middle East and the Jewish World.\n

**$d = 50$**

Melrose Creek is slated for redevelopment in where hundreds of 150-odd residents live in the effort to develop a pedestrian-friendly and quieter downtown in a neighborhood that has been a struggling housing market for their entire lives.\n\nAfter decades of straddling one of the most culturally open places in San Francisco, Melrose Creek's market-driven nature has resulted in popular zoning problems—including a seismic event lifting off of a tent park and opulent vistas sunbathed against the bicycle-friendly

Light Air Rating/Bearing Temperature. Can Handle Winds from Down (1 to 15 ft/min. - 30 It should be able to raise to about 53 mph, 35 mph = 24% duty on stiff and T-shielded loads). Can Handle Medium Dry conditions and Good Condition\n\n\nHybrid Extra Performance." Two Driven In Woofer Lighting Stations! 1967-1992\n\n\nAlso applicable to the best gasket material available. No Gasket Limit forlyst glass.\n\n\n

**$d = 100$**

In the last three years, items were being traded around the world for less in bitcoin. Reuters estimates that its stock price dropped 19%.\n\nYesterday, China has kind of happened. On Yahoo Finance, Rory Schrimpf wrote ("A good amount of Chinese billionaires are planning formal plans for a market of bitcoin for a fund," his Twitter feed says). The gist itself looks like this:\n\nA "bitcoin meeting" that can potentially all but guarantee big deals in bitcoin might come down to "

Even though the story wants to be understood, many people who have experienced hardfought battles of adversity are still going to think that what they experienced in the Battle of Telluride will help define their narratives in a way which will bring this story to the public's attention and invite new dialogue and reconciliation, news organisations have widely concluded that dreams of revolution come easy to ordinary people who make difficult choices. What we often forget is that urbanisation has devastating dams, fragmentation, and denial of services and employment

Table S4: **AriEL samples using GPT-2 as Language Model.** Even for a large language model such the small GPT-2, which has 117M parameters, and over 50K subwords vocabulary, AriEL folding resulted in high quality samples for a wide range of latent dimensions. Here the samples are 100 subword symbols long. This proves the float 32 representation constraint does not represent a limitation at least up to this limit.

## 12   SAMPLES FROM ARIEL TRAINED ON GUESSWHAT?!

|             | 16                                                                                                                                                                                          | 512                                     |
|-------------|-------------------------------------------------------------------------------------------------------------------------------------------------------------------------------------------|-----------------------------------------|
| AriEL       | is it silver ?                                                                                                                                                                             | bowel in white colour ?                 |
|             | is it the trash can ?                                                                                                                                                                     | is it the steel top in white ?          |
|             | look like food ?                                                                                                                                                                         | is it in left ?                         |
| Transformer | foreground am exactly exactly by left same cat dog by                                                                                                                                     | does it the taller ?                    |
|             | bed about bike base base us on ' electricity poster clothing a clothing clothing clothing clothing clothing clothing clothing clothing ?                                                 | is it she is ?                          |
|             | trees stoplight ground ground bird side bikes ?                                                                                                                                          | is it that directly ?                   |
| AE          | a person ?                                                                                                                                                                               | is it a left side ?                     |
|             | the table ?                                                                                                                                                                             | are they in the left side ?             |
|             | is it in the ?                                                                                                                                                                          | a person ?                              |
| VAE         | is the one one of us ?                                                                                                                                                                  | something it to white right side ?      |
|             | the person on a person a ?                                                                                                                                                             | all a person , blue right ?             |
|             | are they in the left of the left side of the ?                                                                                                                                        | all white blue one of front , ?         |

Table S5: **AriEL samples training the Language Model on the real dataset Guess-What?!.** Most samples created through AriEL seem easily interpretable if not fully grammatically correct. Transformer appears to be very difficult to sample from the latent space, and it produces very poor language for $d = 16$, and more reasonable even though hardly interpretable for $d = 512$. AE and VAE seem to produce more reasonable sentences, still often hardly interpretable, and less diverse than AriEL.

