# OpenReview forum: "AriEL: volume coding for sentence generation comparisons"
_ICLR.cc/2022/Conference — ICLR 2022 Submitted_

### Official Review · Reviewer_SHoR · 2021-11-02

**Correctness:** 3
**Technical Novelty And Significance:** 3
**Empirical Novelty And Significance:** 3
**Recommendation:** 6
**Confidence:** 2

**Main Review:**

Strengths:
- A novel approach for latent modelling
- Strong results in terms of validity

- Weakness
In Table 1, the validity percentage of a Transformer with 512 latent dimensionality is only 17.2%. This low score strongly contrasts with our knowledge that a well-trained Transformer language model is very strong at producing valid sentences. One hypothesis is that the amount of training data is not sufficient for Transformer. If this is the case, the proposed method may lose its edge when the training data is abundantly available.

In additional to this point, if the main reason for the Transformer to fail in validity is because the vanilla Transformer was not trained on sampled latent space. We shall at least inject random noise during training to make the Transformer robust to sampling.

**Summary Of The Paper:**

This paper describes a novel methodology of volume coding for encoding and decoding sentences. The algorithm is based on arithmetic coding. The qualitative result reveals that the generations from AriEL to be more valid comparing to other models.






**Summary Of The Review:**

I have reviewed this paper last year, I can see the authors have reorganized the content significantly. I will give a weak acceptance this time and willing to discuss among reviewers.  ​

---

> ### Author Response · Authors · 2021-11-20
> **Reply to reviewer 4**
>
> [Weakness In Table 1...] The PAU and PAB metrics, support the view that Transformers are very strong at producing valid sentences when sampled from the output probability distribution, but our validity metric shows that they are very hard to sample from the latent space! So both observations coexist, and are true at the same time. The good performance on PAU and PAB leads us to think that the size of the model is good for the amount of data.
>
> [In addition to this point...] That is a good point. I guess what comes closest to this idea is the Variational Transformer Network (VTN), where the Transformer encoder produces the mean and variance of a gaussian that is sampled to feed the decoder. My argument would be that in that case we wouldn't be testing Transformer latent organization, but VTN's latent organization.
>
> [I have reviewed this paper...] Thanks a lot!
>
> Thanks for the review!

---

### Official Review · Reviewer_qCdr · 2021-11-03

**Correctness:** 3
**Technical Novelty And Significance:** 3
**Empirical Novelty And Significance:** 1
**Recommendation:** 3
**Confidence:** 4

**Main Review:**

My first concern is that the proposed method essentially is arithmetic coding in multiple dimensions---and the mental abstraction I have in mind is that it is closer to the ZIP file format than an autoencoder, focused very much on perfectly reproducing training data and essentially overfitting it.

My next biggest concern is that I don't think I quite understand the problem this paper is trying to solve in terms of downstream use. For example, an autoencoder's encoder is useful because it can be a way to unsupervisedly obtain continuous representations for discrete string data---those representations could've been tested for their informativeness, e.g., by trying to predict sentence properties or cluster from them. Likewise, an AE's decoder is useful for use as a pretrained generator from latent codes: given a generation task with little data, it may be easier to learn a model that learns to generate latent codes (which are then spelled out by the AE decoder) than one that tries to generate text end-to-end. These kinds of ideas and considerations are missing for me in this document that seems very focused on showing the overfitting and recoverability capabilities of the proposed AriEL. (None of these experiments should be hard to conduct as AriEL through returning centers can be made to "look like" an ordinary autoencoder, I just have a feeling that given its focus on overfitting the training data, the interpolations, while maintaining grammaticality, will not be semantically useful, leading to low performance in these downstream tasks---but I would love to be proven wrong!)
In the same vein, I do not understand why the particular real-world dataset was chosen as opposed to any other collection of strings or datasets that are used by and compared to on previous work.

If that wasn't clear enough, here's a different way to phrase the issue: the evaluations in this paper as far as I can see are only concerned with *unconditional* sample quality, i.e., what do samples from randomly drawn points look like. That is interesting to look at to get a sense of the space, yes, but is ultimately of no relevance to any practical application---because if unconditional samples were truly what we wanted, we would almost certainly be better off just sampling from the base LM that AriEL uses to begin with! Or does this base LM perform worse? If you do want to claim AriELs use in unconditional sampling then that really is the dead simple baseline to outperform in your metrics like validity.

Next, I must admit I do not understand the "biasing" fully. Does it categorically rule out some combinations while leaving others untouched (as Section 3.3 paragraph 3 seems to imply, essentially leading to all strings having probability 0 or p where p is the same for all sentences that have nonzero probability) or is it a weighting device that essentially turns the CFG into a PCFG in which not all strings are equally likely and no string that is possible under the CFG ends up with probability 0 (which is what Figure 1 seems to show)?

Finally, I have doubts about the inductive bias of using dimensions in this round-robin per-word fashion and would like to see whether effects related to that periodicity and subword tokenization appear. One possible experiment to run here to gauge the problem might be to look at where errors occur in data or whether any other patterns arise that pertain to position (early/late/periodical for low dimensionalities)

Other issues or questions:
- the abstract says "We compare to" which sounds like AriEL is an existing technique and not a new contribution of this paper
- Section 1 line 6 says "such type"---what type?
- Section 1 paragraph 3 says "objective benchmark"---what is objective about it and compared to what "subjective" other benchmarks?
- Section 2 line 4 calls volume codes distributed representations but defines the latter as points in R^d rather than values, leading to a contradiction
- Section 2 paragraph 2's citations strike me as old; I am not intimately familiar with advances on these methods since 2017, but would be surprised if there weren't any
- Are the letters in A|B|AA|... in Figure 1 nonterminals or terminals? I suppose they are terminals for the explanation/picture to make sense, but then convention would have them be lower-case instead of upper-case :)
- The last paragraph of section 3.1 mentions RNN-based langauge models, but AriEL should work with any *softmax-based* autoregressive LM as I understand it, including Transformer-based models like the GPT family
- I also admittedly do not understand why there is no linear dependence on vocabulary size in AriEL---lines 9 and 10 in both sides of Algorithm 1 iterate over the vocabulary in a linear way, do they not? (Some of that could be asymptotically recovered with clever sampling, but I think this setting does not lend itself to such asymptotic optimizations, though that might still be good to mention.)
- Unless I missed it, Algorithm 1 is also never referenced in the main text.
- Line 1 on page 4 spells "Transformer" as "Tranformer"...
- ...and it is unclear to me how it is a fixed-length representation unless you cheat by establishing a maximum length and truncating/padding (which you later say you do---this might be good to state upfront to avoid confusion)
- instead of $d_{ff}$ try $d_{\mathit{ff}}$ or something similar to get proper spacing between letters in subscripts
- Testing only 2 or 20 layers is a *huge* difference and even a priori I would expect that both the most commonly chosen and the best-working number of layers is somewhere in the middle
- Section 3.3 paragraph 4 describes "unbiased" sentences as those that go *against* the bias defined (whatever that actually is, see the major confusion above)---shouldn't we then call them something like "opposingly-biased"?
- Generation/Deocding Quality is missing a metric that counts how many of the 10k are grammatical. Yes, if they are all the same sentence and that sentence happens to be correct that measure would give a 100% that looks more impressive than it is, but with that caveat I would rather *know* this number than now know it.
- Relatedly, it took me embarassingly long to understand why validity is the "most important of the metrics" as section 4.1 puts it---may benefit from spelling out that this is indeed the principal metric right where it is defined :)
- I unfortunately also have some quesy feelings about the uniqueness in metrics like validity. For example, if I draw 5 samples, chances are they will all be unique so I can get a high ratio score. Great! But if I draw 5 million samples, it is very unlikely that I will get out 5 million unique sentences, so I will get a low score here. Of course, as will other models, so comparisons are fair, but the fact that as the number of samples tends to infinity the scores tend to 0 strikes me as a sign that the metric definitions aren't fully thought through. If a specific suggestion helps: define validity not as a ratio but as a count of unique sentences and let the number of samples tend to infinity.
- Prediction Quality metrics are ordered in that PAB < BA < GA... is that correct? Might be useful to specify that way :)
- Section 3.4.2 talks about "interpretability," a very loaded technical term---do you mean something like fluency or legibility? Similarly to the choice of "biased" poorly chosen terminology may hurt this paper more than the actual contribution deserves.
- Section 3.4.3: 15 pairs feels like a low number, do estimates truly converge with this few samples? Put differently, is Figure 2 as smooth as it looks? Putting in the actual points may help show just how cleanly sampled that space is.
- Speaking of Figure 2, first, what is the y axis? Validity as defined in the ratio way (see above for my concerns on that) or something new? And what is this lower bound 0.746 that is "related to the language complexity"? Can you spell out that currently somewhat nebulous relationship?
- Section 4.2 paragraph 2: can you quantify "almost all"?

**Summary Of The Paper:**

The paper proposes to use an autoregressive language model's hidden state to index multidimensional arithmetic coding representations of vocabularies and sentences. This way the latent space's geometry is directly connected to the words chosen at the cost of sensible inductive bias between similar sentences. Evaluations of unconditional samples on toy data and small-scale real data are made. While interpolations are presented, the proposed model and the autoencoders it compares to are only ever evaluated for fluency and grammaticality, but not for semantic relatedness.

**Summary Of The Review:**

The paper suffers from a lack of problem that is demonstrably made headway on, i.e., where a proposed method improves over sensible baselines on sensible metrics, largely because data, baseline, and metrics are chosen and justified poorly in my opinion.

That is why I currently cannot see what anyone could get out of this paper as it is (other than remembering that arithmetic coding and k-d trees exist and possibly being tempted to think about volumes in latent spaces and the relation between overfitting and representational capacity a bit more, no thanks to the paper which does not ever talk about these issues itself).

Finally, I should disclose: I have reviewed this paper before (but do not know its authors or anything beyond that) and some more fundamental concerns are unchanged from my last review, though the paper has improved noticeably in my mind (thank you for these improvements, it really has gotten better). My review is thus focused on what I believe is possible to still improve about this paper rather than trying to reiterate previous concerns---that is why I have little to say about the evaluations performed themselves.

---

> ### Author Response · Authors · 2021-11-21
> **Reply to Reviewer 3**
>
> [My first concern is that the proposed method...] The argument we tried to defend is that AriEL is the one that is more able to reconstruct any sequence, inside and outside the training bias. The LM can be prevented from overfitting, as in any standard training procedure.
>
> [My next biggest concern is that...] Our intention was to prove that standard AE, even if by common knowledge should be used as generators, they are factually hard to sample from the latent space. We wanted to propose a method to fix that flaw. All the metrics are performed over held out data, biased and unbiased, with zero overlap with the training data, to make it hard to argue that AriEL was overfitting.
>
> [If that wasn't clear enough...] It is a good idea, but by doing that AriEL would have a decoder with twice the parameters the LM when sampled from its latent space, and we thought that would raise the suspicion in the reader, who would not be able to tell if the advantage came from AriEL, or from having twice the amount of parameters.
>
> [Next, I must admit I do...] We modified the caption of Figure 1 to make clear that it is a simple example to convey the idea quickly. The biasing is closer to the first option you describe, even though it's done at the word level, so, it won't result in equal probability to every sentence.
>
> [Finally, I have doubts about the...] Yes, that is definitely an interesting direction to explore. Unfortunately we didn't have time to design and run the experiments to satisfactorily reply to this question in the rebuttal period.
>
> Other issues or questions:
>
> [the abstract says...] done!
>
> [Section 1 line 6 says...] done!
>
> [Section 1 paragraph 3...] done!
>
> [Section 2 line 4 calls...] Volume codes are a subset of distributed codes, since you need a collection of adjacent points to make a volume code, while one point suffices to make a distributed code. We reworded it.
>
> [Section 2 paragraph 2's...] done!
>
> [Are the letters in A|B|AA|... in Figure 1...] change done!
>
> [The last paragraph of section 3.1...] change done! There is a way to make the encoder with matrices multiplications if the representations of all the characters are available, therefore getting rid of a for loop, but since we didn't explain it in the paper, I can remove the 'RNN-based' mention.
>
> [I also admittedly do not...] change done!
>
> [Unless I missed it...] It is mentioned in section 3.1.
>
> [Line 1 on page 4...] Thanks!
>
> [...and it is unclear to me...] Ok! but we mention it in the second sentence about Transformer, not very late!
>
> [instead of ff try ff...] DONE
>
> [Testing only 2 or 20 layers...] That is true, but for the other models we only changed the latent dimension, so it felt that it would look more suspicious if we changed several hyper-parameters.
>
> [Section 3.3 paragraph 4...] It initially had sentences from both, bias en opposingly biased, but then we chose to remove all biased, to make the metric more clear. It is still a set way larger than the biased set, since an unbiased sentence can have all the word pairs right except for one, and that will make it directly unbiased. I understand your criticism though.
>
> [Generation/Deocding Quality...] It is a good point, but we had to remove other metrics of interest to make the table fit the article width, and the grammaticality without uniqueness has the drawback you mention, so we felt it could be left aside.
>
> [Relatedly, it took me embarassingly...] DONE
>
> [I unfortunately also...] We work with a finite amount of samples, for a grammar that can produce millions of sentences (SM2), and the metric tries to quantify how easy is to produce different high quality sentences. A ratio gives a quick impression of how far from the maximum we are. If a count was used, the reader would have to keep in mind the number of samples in the test set, and it was our impression that that would make the read more obscure.
>
> [Prediction Quality...] We changed the description of the metric, but with the previous definition you were right.
>
> [Section 3.4.2 talks about "interpretability,"...] An article on grounded dialogues called the Wizard of Wikipedia uses engagingness scores, and asks surveyees to score from 1 to 5 how much they liked the sentences. That sounded very vague to us. We thought interpretability would make it easy to explain to the surveyees, without guiding them into giving the reply that we hoped them to give. It seemed like a concept that could be quickly grasped by anybody not necessarily in the NLP community.
>
> [Section 3.4.3: 15 pairs...] We are rerunning the code for the plot to show it as you suggested.
>
> [Speaking of Figure 2...] Yes, same validity definition as in the toy dataset, as defined in section 3.4.3. We changed the caption of figure 2.
>
> [Section 4.2 paragraph 2...] Changed wording and placed the actual number.
>
> [The paper suffers ...] We motivated more the intro, we restructured and detailed the algorithm, improved the motivation of the metrics.
>
> Thanks a lot for the detailed feedback!

---

> > ### Comment · Reviewer_qCdr · 2021-11-22
> > **Replying to some points**
> >
> > [My first concern is that the proposed method...] Still, Ariel is closer to a "LM interface" than an actual model like autoencoders and Transformers are. Is that a wrong impression?
> >
> > [My next biggest concern is that...] My point is that Ariel doesn't "fix a flaw" so much as "do something completely different"---throwing the baby (i.e., actual usefulness) out with the bath water.
> >
> > [If that wasn't clear enough...] I don't understand your reply here. Put simply: can the base LM not also be evaluated on the metrics with the exact same tasks? If anything, that baseline should have *fewer* parameters.
> >
> > [Next, I must admit I do...] I still feel as confused.
> >
> > [Testing only 2 or 20 layers...] But you already change them! What I am asking is to change them *sensibly* and not only evaluate extremes.

---

### Official Review · Reviewer_JWsh · 2021-11-03

**Correctness:** 2
**Technical Novelty And Significance:** 1
**Empirical Novelty And Significance:** Not applicable
**Recommendation:** 3
**Confidence:** 3

**Main Review:**

1. I feel the paper is poorly-organized. I repeat like 5-6 times to get what it's trying to do in introduction. This could immediately be a reason to reject the paper as I think it's not gonna benefit audience in ICLR.

2. After digesting it, I think I like to problem this paper discussed, and it's very pertinent to the representation learning. However, I am not sure if I got 100% understanding of the content as it's again poorly presented. Leaving all the materials in an algorithm table without much explanations make it hard to understand. I think a running example should be provided to make reader easier to understand what's going on. More importantly, authors seem to assume readers know what AC is but honestly I don't see the necessity.

3. From the start to end I am not sure why Volume Code is good. This isn't well motivated. And I honestly don't know how the metric is picked. I don't see these properties are necessarily good. If my VAE or other models are learning a representation toward a specific task then why it's necessary to follow these designs?

4. I feel all the comparison models can be used as language models. So my understanding is that Ariel can be used to generate code provided an underlying LM. Why not train VAE or other stuff in LM task and then apply Ariel to see the results?

5. Following 4, for this type of paper, an observation is provided and usually it should also propose a way to augment the existing methods to match the proposed metric. So it reads to me that authors should propose a way to show that representation learned by VAE can be improved by adding some components. And in my point 4, I do feel treating it as a LM can be tested.



**Summary Of The Paper:**

The paper proposed a method to encode/decode a sentence (or say a sequence of tokens) into/from a volume instead of just a point(vector) in d-dimensional. The paper shows some good attributes of this method via some experiments.

**Summary Of The Review:**

Overall, I think it's not a presented in a straightforward way and it's lack of many details for readers to fully digest it. I feel the topic is interesting but in practice I don't know why it's important and author didn't motivate this. I will be considering to raise the score if the introduction is re-written in a motivating form.

---

> ### Author Response · Authors · 2021-11-21
> **Reply to reviewer 2**
>
> [I feel the paper is poorly-organized...] Sorry, there was an important sentence missing that was commented. We worked on the introduction to make it more motivating as well.
>
> [After digesting it, I think I like the problem...] We reworked on the algorithm presentation as well, section 3.1. We added a running example. We added a sentence in Related Work, AC paragraph, to clarify AC compression.
>
> [From start to end I am not sure why Volume Code is good...] Hopefully the new introduction should provide a better motivation. Metrics are chosen to take advantage of the new possibilities opened by the CFG, that allows us to check the grammatical correctness of the language produced, and the possibilities opened by the random bias, that allows us to automatically check if the bias is respected. Standard metrics such as perplexity, would only check if the next word chosen by the model matches in the exact time step the word in a reference sentence. In our view, that makes it an obscure metric for language quality, used in the lack of a better one for (non CFG) human language. We added this clarification in section 3.4.1.
>
> [I feel all the comparison models can be...] It is a good idea, but by doing that AriEL would have a decoder with twice the parameters the LM when sampled from its latent space, and we thought that would raise the suspicion in the reader, who would not be able to tell if the advantage came from AriEL, or from having twice the amount of parameters.
>
> [Following 4, for this type of paper, an observation...] One of the ideas we wanted to convey is what you mention, to explain how any LM can be used in a new way, a way that bounds their representations, and makes them volumetric, making it easier to retrieve any pattern, those that are in the training set, and those that are not. We modified the introduction to help the reader get a clearer picture. Another way to answer to your comment is by seeing that VAE is already an iteration over AE, so we show how our iteration over AE surpasses VAE.
>
> [Overall, I think it's not a presented...] We added a motivating introduction to reply to your concerns. We explained in more detail and restructured the description of the algorithm. We motivated better the choice of metrics in section 3.4.1.
>
> Thanks a lot for the feedback.

---

### Official Review · Reviewer_JLnX · 2021-11-04

**Correctness:** 2
**Technical Novelty And Significance:** 2
**Empirical Novelty And Significance:** 2
**Recommendation:** 3
**Confidence:** 3

**Main Review:**

1. The writing of this paper has much room for improvement. For example, the description of the method part in Sec 3.1 is super unclear. It's hard to fully understand the method without referring to the pseudo-code. Some of the key operations (e.g., "find function") are moved to the appendix. And the notations in the pseudo-code are also confusing, e.g., a word token is sometimes used with or without a subscript -- Inconsistent notations of "(s_j)_{j<i}" and "s > s'" seem to denote previous words before the i-th word.
2. Conceptually it's still not clear why splitting the latent space using the LM (trained on biased text data) could be beneficial to the generalization for unbiased text. Even though the coding method could efficiently utilize its latent space to fill sentences in all its d dimensions, wouldn't it also cover the bias from the LM trained on biased text data?
3. I don't see most of the evaluation metrics used in this paper in the literature. I am a bit skeptical whether these metrics are convincing enough to evaluate these encoding methods on more realistic, complex datasets in addition to the toy & narrow-domain datasets used in this paper.
4. The other models in comparison seem to be artificially tuned in some ways which lead to pretty low performances. For example, using only 2-layers of encoder & decoder or a small dimension for Transformer, sampling the latent space of Transformer. It's not clear how the proposed method compared with stronger models.

**Summary Of The Paper:**

This paper introduces an entropic coding method that is able to compress a sentence into a latent space and perform text generation through a uniform sampling from the implicit latent space. Experiments are conducted on a toy dataset and a narrow domain text dataset to show the effectiveness of the proposed method w.r.t. AE, VAE, and Transformer models.


**Summary Of The Review:**

I would suggest a rejection of this paper given the current draft.

---

> ### Author Response · Authors · 2021-11-21
> **Reply to Reviewer 1**
>
> [1. The writing of this paper...] We restructured the explanation of Sec 3.1 to improve the clarity, as you asked, and we added some explanations at the beginning of section 3.1 to make the notation more transparent. "(s_j)_{j<i}" denotes a sequence, as in maths rounded parenthesis are used to denote sequences, s_j denotes observed sample, s and s' denote the words as random variables. The find operation takes a lot of space to explain and in terms of content doesn't add much, that is why we thought it would be better to keep it in the appendix.
>
> [2. Conceptually it's still not clear...] The idea we tried to convey through the reconstruction study, is that, AriEL allows to keep a representation for any sequence in that bounded domain, while other techniques, seem not to have a representation for unseed data at all. Our argument is therefore that keeping a bounded representation for unseen/test data, makes their representations easily accessible, which will ease generalization.
>
> [3. I don't see most of the evaluation metrics...] We added a comment at the beginning of section 3.4.1. Given the possibilities to check the grammar and the 'semantic' bias introduced by using a CFG, we can look more closely at the quality of language produced, than with standard metrics. Those notions are not easily quantifiable with human datasets, nor standard metrics such as perplexity.
>
> [4. The other models in comparison seem...] We added a paragraph at the end of section 3.2. All of them are sampled from the latent space, not only Transformer, and all of them are constrained to a small latent space, not only Transformer. All of them except for AriEL are 2-layers of encoder & decoder, because they improved performance compared to when we run them with only 1 layer. All of them have about the same number of parameters. These choices were made to make it difficult to argue that the set up benefitted one method over another.
>
> Thanks a lot for the feedback!

---

### Decision · Program_Chairs · 2022-01-20

**Decision:**

Reject

**Comment:**

The paper proposes an entropic coding approach for sentence embedding.

Reviewers have spent good efforts in reviewing. They generally feel the problem is important/interesting, but also found it difficult to understand the paper. Thus, the authors are encouraged to thoroughly revise the paper according to the reviews provided, and another round of review is needed to better determine the merits of this paper.